# Effects of Environmental Cadmium Exposure Sufficient to Induce Renal Tubular Dysfunction on Bone Mineral Density Among Female Farmers in Cadmium-Polluted Areas in Northern Japan

**DOI:** 10.3390/toxics13080688

**Published:** 2025-08-18

**Authors:** Hyogo Horiguchi, Etsuko Oguma, Kayoko Miyamoto, Yoko Hosoi, Fujio Kayama

**Affiliations:** 1Department of Hygiene, Kitasato University School of Medicine, Sagamihara 252-0374, Japan; oguma@med.kitasato-u.ac.jp; 2Department of Environmental and Preventive Medicine, School of Medicine, Jichi Medical University, Shimotsuke-shi 329-0498, Japan; kayokomy@nifty.com (K.M.); yk_hosoi2005@yahoo.co.jp (Y.H.); kayamafujio@gmail.com (F.K.)

**Keywords:** cadmium, farmers, bone, renal tubular dysfunction

## Abstract

In the Japanese Multi-Centered Environmental Toxicant Study (JMETS) conducted in five areas across Japan, we demonstrated that bone mineral density (BMD) in female farmers without renal tubular dysfunction was not adversely affected by exposure to low to moderate levels of cadmium (Cd). We then expanded JMETS to the most Cd-polluted area in northern Japan, Akita prefecture, with area A as the control and areas B and C as Cd-polluted areas (Cd exposure levels: B < C), which also covered more female farmers with a wider age range (20–82 years) and Cd exposure sufficient to induce renal tubular dysfunction. We selected 1267 eligible subjects in the three areas and classified them by age and menstrual status. The distribution of blood and urinary Cd levels over the areas was A < B < C (blood Cd: 2.10, 3.78, and 3.39 µg/L, and urinary Cd: 3.02, 4.29, and 6.15 µg/g cr., respectively; *p* < 0.05), with the steepest age-dependent increase in area C, particularly in older postmenopausal subjects with a urinary Cd level around the threshold for renal tubular dysfunction. Urinary α1-microglobulin (α1MG) and ß2-microglobulin (ß2MG) levels in the three areas also showed age-dependent increases, with higher levels being observed in areas B and C than in area A. Furthermore, ß2MG levels in older postmenopausal subjects were significantly higher in area C than in area A (273 and 157 μg/g cr., respectively, *p* < 0.05). Age-dependent decreases in BMD were noted in all areas, with rapid reductions from peri- to postmenopausal subjects; however, marked differences in each age class were not observed among the three areas. In multiple regression models of BMD in all subjects using age, body weight, grip, urinary creatinine, urinary α1MG or ß2MG, and blood or urinary Cd as independent variables, urinary α1MG and ß2MG levels correlated with BMD, whereas blood and urinary Cd levels did not. Moreover, age and body weight correlated more strongly with BMD than blood and urinary Cd levels. Therefore, Cd, not only at a low level but also at a level that was sufficient to deteriorate renal tubular function, did not affect bones. These results provide further support for Cd exposure itself not directly affecting bones.

## 1. Introduction

The heavy metal cadmium (Cd) widely exists in the environment, and people are exposed to it daily, mainly through food [1]. It accumulates in the human body in an age-dependent manner, primarily in the kidneys, because of its long biological half-life (10–30 years). When the accumulation of Cd exceeds the threshold due to excessive intake, which is often observed with local environmental Cd pollution, multiple proximal renal tubular dysfunction, called Fanconi’s syndrome, develops and is referred to as “Cd nephropathy”. Cd nephropathy is characterized by an increase in the urinary excretion of water, low-molecular-weight proteins, phosphate, calcium, glucose, and bicarbonate ions, which ultimately leads to the development of osteomalacia. In the Jinzu River basin in Toyama prefecture, Japan, which was heavily contaminated by Cd from an upstream mine, more than 200 cases of osteomalacia occurred among local female farmers, who developed pain due to multiple bone fractures, called “*itai-itai* disease” (*itai* means “ouch”) [2].

The main cause of Cd-induced osteomalacia, observed in *itai-itai* disease, is phosphate deficiency derived from the continuous urinary loss of phosphate due to deterioration of the reabsorption function of renal tubules, which is clinically recognized as a low serum level of phosphate [3]. Therefore, bone injury in *itai-itai* disease develops secondary to Cd nephropathy, and not as a result of the direct effects of Cd on bone tissues. In the Japanese Multi-Centered Environmental Toxicant Study (JMETS) [4], in which the relationship between Cd exposure and health was investigated in female farmers in five areas of Japan, we demonstrated that bone mineral density (BMD) was not affected by exposure to low to moderate levels of Cd, which were insufficient to induce renal tubular dysfunction [5]. However, other studies conducted in various countries indicated the direct effects of Cd, at a very low level that was inadequate to cause renal dysfunction, on bone that ultimately led to osteoporosis [6,7,8,9,10]. While this discrepancy may be partly attributed to differences in races, ethnicity, dietary habits, and the methods used to measure BMD, the marked difference in the effects of Cd on bones between these studies and JMETS suggests the presence of other contributing factors. Therefore, populations exposed to a wide range of Cd levels, from a very low level to a level sufficient to induce renal tubular dysfunction, need to be comprehensively examined in order to compare the effects of Cd on bones in consideration of the involvement of renal tubular dysfunction between Cd exposure levels.

Among the five areas covered by JMETS described above, the highest Cd level was observed in Odate city in Akita prefecture, located in the northern part of Japan [4,5]; however, this level did not affect renal function in female farmers in their 40s–60s. We then expanded the investigation in Odate city to target more female farmers in their 20s–70s, and we also examined adjacent areas in Akita prefecture with higher Cd pollution levels than Odate city, namely, Kazuno city and Kosaka town, in which there were female farmers in their 40s–80s [11,12]. In the expanded JMETS, we expected elderly female farmers, particularly those in their 70s, to have renal tubular dysfunction as well as exposure to high levels of Cd. Therefore, in the present study, we comprehensively investigated the relationship between blood and urinary Cd levels and BMD in female farmers exposed to a wide range of Cd levels, with a high level being sufficient to induce renal tubular dysfunction, in order to clarify intrinsic differences in the effects of various Cd exposure levels on bones.

## 2. Materials and Methods

### 2.1. Study Areas and Populations

The Committee on Medical Ethics of Jichi Medical University approved the research protocol, and this study was conducted in accordance with the ethical standards laid down in the Declaration of Helsinki. The present study consisted of populations from 3 farming areas in Akita prefecture: Happo town and Gojome town as a control (A), Odate city (B), and Kazuno city and Kosaka town (C) [11]. There were large mines and their affiliated smelters in area C, from which the Yoneshiro River runs through area B to the west. The rice fields in areas B and C were contaminated by mine-derived Cd through irrigation from river water, and also through air pollution from the smelters in area C. Therefore, Cd pollution levels were expected to be higher in area C than in area B. The 2 towns in area A are located by the Sea of Japan, on the opposite side of areas B and C in Akita prefecture, and away from the Yoneshiro River; therefore, there was no obvious Cd contamination from the mines. Areas examined in JMETS other than Akita prefecture were not included to eliminate unknown effects that may be attributed to differences in the areas across Japan.

Female farmers who consumed home-harvested rice in the winters of 2006, 2001–2002, and 2003–2004 were investigated in areas A, B, and C, respectively. With the cooperation of the local Japan Agricultural Cooperative (JA) and municipalities, female farmers aged 20 years and older were recruited from every farming hamlet for health examinations. There were 4862 and 4446 JA member households in each of the 2 JAs in area A, 6749 in area B, and 4187 in area C, from which 291, 938, and 520 applicants were recruited, respectively. In total, 242, 725, and 438 applicants in areas A, B, and C, respectively, participated in the health examinations. Exclusion criteria were as follows: ex- and current smokers, the presence of chronic renal failure, nephritis, renal tumors, rheumatoid arthritis, systemic lupus erythematosus, sarcoidosis, hyperthyroidism, steroid hormone therapy, an insufficient volume of blood sampled, incomplete answers in questionnaires, and less than a 10-year history of eating locally harvested rice. Accordingly, 220, 655, and 392 subjects in areas A, B, and C, respectively, totaling 1267 subjects, were ultimately analyzed.

### 2.2. Procedures for Health Examinations

Orientation sessions on the purpose of the present study and its protocol were held approximately one week before health examinations, and written informed consent was obtained from all subjects. In these sessions, subjects were handed a self-administered questionnaire on their place of residence, the origin of rice consumed, medical history, including diseases and therapies, smoking habits, and lifestyles, which was to be completed by the health examination. In the health examination, peripheral blood and second-morning urine samples before breakfast were collected, and the weight, height, grip strength, and BMD of subjects were measured. Nurses checked and collected the questionnaire. Body mass index (BMI) was calculated by dividing weight (kg) by height (m) squared. The grip strength of the non-dominant hand was measured with a hand dynamometer three times, and the highest value was adopted as an indicator of physical activity. BMD was measured by dual energy X-ray absorption (DXA) on the non-dominant forearm with a DTX-200 (Osteometer MediTech, Inc, Signal Hill, CA, USA), which scanned DXA at distal sites of the radius and ulna between the 8- and 24-mm points. The percentage of the young adult mean (20–44 years old) (YAM%) was also used as an indicator of the fragility of bones, with BMD < 80% of the YAM being classified as “decreased BMD” according to the criteria of the *Journal of Japan Osteoporosis Society* (Japanese 2015 guidelines for the prevention and treatment of osteoporosis).

### 2.3. Analyses of Blood and Urine Samples

Whole blood samples were prepared with heparin for Cd measurements, and serum samples were prepared without anticoagulants by centrifugation for biochemical measurements. Urine samples were placed on ice immediately after collection and subjected to three treatments: the addition of one drop of 20% sodium carbonate to prevent the destruction of ß2-microglobulin (ß2MG) due to low pH [13], the addition of one drop of 0.1 mol nitric acid to stabilize Cd, and no additives for other measurements. Urinary α1-microglobulin (α1MG) and ß2MG as well as serum and urinary creatinine (Cr) levels were measured using a latex agglutination method and the Jaffe reaction method, respectively. Luteinizing hormone (LH), bone-specific alkaline phosphatase (BAP), and bone Gla protein levels in serum were assessed by an immunoradiometric assay (IRMA), ELISA, and IRMA, respectively, and N-telopeptide crosslinked collagen type 1 (NTx) and deoxypyridinoline (D-Pyr) in urine by ELISA. Mitsubishi Kagaku Bio-Clinical Laboratories, Inc. (Tokyo, Japan) conducted all biochemical measurements. The estimated glomerular filtration rate (eGFR) was calculated as an indicator of renal glomerular function, using the equation for Japanese females: eGFR (mL/min/1.73 m^2^) = 194 × serum Cr − 1.094 × age − 0.287 × 0.739 [14].

### 2.4. Measurement of Cd Concentrations in Whole Blood and Urine

The methods used to measure Cd concentrations in whole blood and urine were previously described [11]. Briefly, blood samples were decomposed by microwaving with nitric acid, and Cd concentrations were measured using HP4500 ICP-MS (Yokogawa Analytical Systems, Tokyo, Japan). Urine samples were mixed with nitric acid, left to stand for 24 h, and Cd concentrations were then measured using flameless atomic absorption spectrometry (SIMAA 6000; Perkin Elmer, Norwalk, CT, USA). Ultrapur nitric acid (Kanto Kagaku, Tokyo, Japan) and nitric acid for the Ultratrace Analysis (Wako Pure Chemical Industries, Osaka, Japan) were used for measurements. Indium and thallium were added to all samples as internal standards. The standard solutions for Cd, indium, and thallium were purchased from Wako Pure Chemical Industries. The effects of the sample matrix on Cd measurements were examined using standards, including CRM195 (Institute for Reference Materials and Measurements) for peripheral blood and 69,071 Level 1 (human urine) (Bio Rad) for urine. The limit of quantitation (LOQ) was calculated at 10× the standard deviation (SD) of reagent blank measurements (n = 5). IDEA Consultants, Inc. (Metocean Environment Inc., Shizuoka, Japan) conducted the measurements of Cd concentrations.

### 2.5. Statistical Analysis

Values less than the LOQ or the minimal measurable limit were replaced with half the value for the statistical calculation. Data that followed a normal distribution, judged by their normal probability plots, were presented as arithmetic means with SD, and the others as medians with 25th and 75th percentiles. The values of urinary concentrations were adjusted by the urinary Cr concentration and presented as “µg/g cr.”. When we compared groups classified by age or urinary Cd levels, the Bonferroni test or Bonferroni–Holm test (for more than 4 classes) was used for normally distributed data, and the Steel–Dwass test for data that did not follow a normal distribution. Changes in groups classified by age and urinary Cd were examined by the Jonckheere–Terpstra test. In multiple regression analyses, data obtained on urinary α1MG and β2MG, blood and urinary Cd, and urinary Cr were converted into base-10 logarithms. Urinary variables were not adjusted by the Cr level, which was instead added as an independent variable [15]. Several multiple regression models were used to avoid multicollinearity between variables with strong correlations, such as urinary α1MG and β2MG and blood and urinary Cd. Furthermore, the multicollinearity of variables was confirmed by variance inflation factors (VIFs). The adequacy of multiple regression models was assessed by multiple correlation coefficients and a normal probability plot of the residuals. The significance of factors was judged by their *p*-values and partial correlation coefficients (β). Statistical analyses were performed using SPSS release 27.0 (SPSS Japan, Tokyo, Japan) based on the basic management of data by Mac Excel Tokei ver. 3.0 (Esumi, Tokyo, Japan).

## 3. Results

### 3.1. Grouping of Subjects

The subjects selected in each area were divided into four classes according to age and menstrual status: premenopausal (33–48 years old), perimenopausal (49–55), younger postmenopausal (56–65), and older postmenopausal (66–82) (Table 1). There were no menopausal subjects in the premenopausal class, while all subjects were postmenopausal in the younger and older postmenopausal classes. The perimenopausal class included subjects with and without menopause. Since there were no subjects younger than 33 years in areas A and C, 22 subjects aged 20–32 years in area B were omitted from subsequent analyses of the three areas (total number, 1245).

The four classes did not show any significant differences in mean age among the areas. In addition, serum LH levels in the premenopausal classes were low and rapidly increased from the premenopausal to perimenopausal classes, followed by high levels being maintained in the postmenopausal classes in all areas, reflecting increases in LH secretion in response to the decrease in internal estrogen levels due to menopause. Therefore, this grouping allowed us to simultaneously adjust for the effects of two major bone-relevant factors, age and menstrual status, and examine the intrinsic effects of Cd on bones.

### 3.2. Cd Levels in Peripheral Blood and Urine

Then, the levels of Cd that accumulated in subjects in the three areas were assessed using blood and urine Cd concentrations (Table 2, Figure 1 and Figure 2). In area A, no significant differences were observed in blood Cd levels among the age classes, whereas urinary Cd levels were significantly higher in the postmenopausal classes than in the premenopausal classes. In contrast, blood and urinary Cd levels were significantly higher in most of the age classes in areas B and C than in area A and also showed age-dependent increases. In addition, urinary Cd levels in the peri- and postmenopausal classes were higher in area C than in area B, with differences in areas A and B becoming more pronounced with aging. The urinary Cd level in area C was as high as >8 µg/g cr., which was 2.7-fold higher than that in area A in the older postmenopausal class. Furthermore, subjects with urinary Cd levels > 10 μg/g cr., the traditional threshold for Cd-induced renal tubular dysfunction [16], accounted for 29.1% of the older postmenopausal class in area C, and 7.6 and 0% in areas A and B, respectively. These results confirmed that the order of Cd exposure levels was as follows: area A < area B < area C, and the exposure level of Cd in area C, particularly in elderly subjects, was sufficient to induce renal tubular dysfunction.

### 3.3. Urinary α1MG and ß2MG Levels and eGFR

Therefore, the deterioration of renal function in subjects exposed to high levels of Cd was investigated. In subjects in the three areas, renal tubular function was assessed by urinary α1MG and ß2MG, and renal glomerular function by eGFR, both of which are closely related to bones (Table 3, Figure 3 and Figure 4). All age classes in the three areas showed significant age-dependent increases in urinary α1MG and ß2MG levels, with more pronounced increases being observed in areas B and C than in area A. Urinary ß2MG, which is more sensitive than urinary α1MG, in the older postmenopausal class was significantly higher (approximately 2-fold) in area C than in area A. It is important to note that there was one subject with Cd nephropathy in the older postmenopausal class in area C (blood Cd 31.2 µg/L, urinary Cd 18.8 μg/g cr., and urinary ß2MG 15,300 μg/g cr.) [11]. On the other hand, no significant differences were observed in eGFR in each age class among the three areas, while an age-dependent decrease was noted in every area. These results indicate that area C included subjects with renal tubular dysfunction induced by Cd exposure.

### 3.4. Body Weight and Grip Strength

In addition to Cd accumulation and renal tubular function, body weight and grip strength were measured in subjects to evaluate the effects of Cd on bones, as they are relevant to bone strength (Table 4). Body weight showed an age-dependent decrease in each area, and there were no significant differences between the three areas. BMI, an indicator of obesity, was also measured and found to be similar in all age classes in the three areas. Grip strength also showed an age-dependent decrease, with no significant differences among the three areas, similar to body weight. The results showing age-dependent decreases in body weight and grip strength suggest that these factors need to be considered when investigating the effects of Cd on bones.

### 3.5. BMD and YAM%

Then, BMD and YAM% were examined in age-classified subjects in the three areas (Table 5). BMD and YAM% both showed age-dependent decreases in each area, with a rapid reduction from the peri- to postmenopausal classes to approximately 80% of the YAM, the level of “decreased BMD”. However, no significant differences were detected in each age class among the three areas.

These decreases in BMD and YAM% were very similar to the results obtained for body weight and grip strength (Table 4, Figure 5), suggesting their effects on bones. These patterns also resembled that of the age-dependent deterioration of renal tubular function, as indicated by increases in urinary α1MG and ß2MG, except for the stronger effects on renal tubular function in elderly subjects in areas B and C. Unlike body weight, grip strength, and urinary α1MG and ß2MG, the patterns of BMD and YAM% were not consistent with those of blood and urinary Cd levels: subjects in areas B and C that accumulated a high level of Cd did not show any decrease in BMD or YAM% (Table 2).

### 3.6. Bone Metabolism

In addition to BMD and YAM%, factors related to bone metabolism, including serum calcium and phosphate and urinary calcium, were investigated, as well as the following bone metabolism markers: bone alkaline phosphatase and osteocalcin as osteogenic markers, and urinary NTx and D-Pyr as osteoclastic markers (Table 6 and Table 7). Serum calcium and phosphate levels were slightly lower in areas B and C than in area A. No significant differences were observed in urinary calcium among the three areas, with rapid increases being detected from the pre- to perimenopausal classes, potentially reflecting the menopause-related acceleration of bone absorption. The four types of bone metabolism markers also showed increases from the pre- to perimenopausal classes, with no significant differences among the three areas, except for urinary NTx in area B.

### 3.7. Age- and Urinary Cd-Classified Analyses

To examine the effects of Cd exposure on bones in more detail, the total number of subjects classified by age and urinary Cd was analyzed. All subjects (n = 1245, 22 subjects aged 20–32 years in area B were excluded) were initially divided using the median and 25th and 75th percentiles of urinary Cd (3.0, 4.5, and 6.5 µg/g cr., respectively), and then further divided by age and menstrual status, as described above. This classification provided four classes of gradual increases in blood and urinary Cd with the same age means in each age class (Table 8), thereby allowing the effects of Cd exposure itself on renal tubular function and bones to be assessed.

Urinary α1MG and ß2MG showed significant age-dependent increases in all urinary Cd classes (Table 9), indicating their natural deterioration independent of Cd exposure. While no significant differences were observed in urinary α1MG or ß2MG among the urinary Cd classes in all age classes, significant increases along with elevations in urinary Cd were noted in urinary α1MG in the peri- and older postmenopausal classes and in urinary ß2MG in the peri- and younger and older postmenopausal classes. These results suggest that renal tubular function was affected by Cd exposure in addition to its natural age-dependent deterioration in elderly subjects.

BMD and YAM% both showed age-dependent decreases, with significant and rapid reductions being observed from the peri- to postmenopausal classes in all urinary Cd classes (Table 9). No significant differences were observed in BMD or YAM% among the urinary Cd classes in all age classes, whereas in the older postmenopausal classes, they significantly decreased along with increases in urinary Cd. These results on BMD and YAM% after adjustments for urinary Cd revealed the mild adverse effect of Cd on bones, which was not detectable in the comparison among the three areas.

### 3.8. Multiple Regression Analyses of Renal Tubular Function and BMD

The observed effects of Cd on bones may be confounded by a number of factors, such as age, body weight, and grip strength, in parallel with BMD and may also be affected by the Cd-induced deterioration of renal tubular function. Therefore, these effects were confirmed using multiple regression analyses of BMD.

To examine the net effects of Cd on renal tubular function, multiple regression analyses of renal tubular function and urinary α1MG and ß2MG were performed using age, urinary Cr, and blood or urinary Cd as independent variables (Table 10). In consideration of multicollinearity between blood and urinary Cd, two models each for urinary α1MG and ß2MG were used. The results obtained indicated that all models had multiple significant correlation coefficients, residuals that were normally distributed, and low VIF values in all independent variables. In all models, blood and urinary Cd correlated with urinary α1MG and ß2MG, although their ß values were lower than those of age.

Then, multiple regression analyses of BMD were conducted using age, body weight, grip strength, urinary Cr, urinary α1MG or ß2MG, and blood or urinary Cd as independent variables (Table 11). In consideration of multicollinearity between urinary α1MG and ß2MG and also between blood and urinary Cd, four types of models were used. All models had multiple significant correlation coefficients, residuals that were normally distributed, and low VIF values in all independent variables. In all models, urinary α1MG and ß2MG correlated with BMD, whereas blood and urinary Cd did not, although their ß values were small, and age and body weight had markedly larger ß values than blood and urinary Cd in addition to their correlations with BMD.

## 4. Discussion

The present study investigated female farmers in three areas in Akita prefecture, located in the northern part of Japan. The control area had no obvious Cd pollution (A), while in the two Cd-polluted areas, one showed a moderate level of Cd exposure without any obvious adverse effect on renal tubular function in local residents (B), and the Cd exposure level in residents in the other was sufficient to induce renal tubular dysfunction (C). We compared the three areas and demonstrated that the Cd exposure levels in the subjects were in the order of areas A < B < C and also increased in an age-dependent manner. Moreover, renal tubular function deteriorated in older postmenopausal subjects in area C, whereas no significant differences were observed in BMD or indicators for bone metabolism among the three areas. In addition to the comparison among the three areas, we combined subjects from all areas and classified them by urinary Cd levels to examine the effects of Cd exposure on bones. The results obtained revealed urinary Cd-dependent decreases in BMD along with the deterioration of renal tubular function in the older postmenopausal class. However, multiple regression analyses did not detect the net effects of Cd on BMD, excluding the effects of confounding factors, such as age, weight, grip strength, and urinary α1MG or ß2MG. These results indicate that not only a low level of Cd but even excessive Cd exposure, inducing the deterioration of renal tubular function, did not exert intrinsic adverse effects on bones, bolstering our previous findings showing that BMD was not affected by Cd exposure at a level that was insufficient to induce renal tubular dysfunction [5]. Therefore, Cd exposure at a low level that does not affect renal tubular function does not appear to directly injure bones.

Based on a large number of observations of individuals living in several Cd-polluted areas in Japan, including patients with *itai-itai* disease, a natural history for the clinical manifestations of chronic Cd toxicity has been proposed. Cd nephropathy and *itai-itai* disease are major two-stage clinical manifestations; however, there are also minor phases in the course of chronic Cd toxicity that show unidirectional and continuous progression [17,18]. Starting with the accumulation of low levels of Cd, the first phase is the excessive accumulation of Cd in the body, reaching high blood or urinary Cd levels without any effects on renal tubular function. Among individuals with excessive Cd accumulation, some develop renal tubular dysfunction with aging, which is the phase of Cd nephropathy. During the long-term and continued loss of phosphate, calcium, and bicarbonate ions into urine due to a disturbance in renal tubular reabsorption, a bone metabolism disorder gradually progresses as a result of phosphorus deficiency and metabolic acidosis, leading to the demineralization of bones. Osteomalacia develops in the final phase, i.e., *itai-itai* disease. The progression of bone injury toward the final phase is often accompanied by renal anemia due to the insufficient renal production of erythropoietin. Specifically, this continuous, unidirectional developing series of phases is referred to as “a spectrum of chronic Cd toxicity”. The subjects analyzed in the present study ranked from no excessive Cd accumulation to early Cd nephropathy without bone metabolism disorders and showed no adverse effects of Cd on bones, which is in accordance with the order of the spectrum.

Therefore, there is a serious discrepancy between the spectrum based on the present results and previous findings [6,7,8,9,10], which demonstrates the adverse effects of Cd on bones at very low levels, even below the Cd accumulation phase in the spectrum. Studies from Western countries, such as Sweden, Belgium, and the U.S., reported the adverse effects of blood Cd < 1 µg/L or urinary Cd < 1 µg/g cr. on bones, while studies from Asian countries detected these effects at <3 µg/L or <3 µg/g cr., respectively. These reports are not in line with the spectrum. On the contrary, the present study did not detect any adverse effects of Cd on BMD even at the highest Cd exposure class, where blood Cd was 5.5 µg/L and urinary Cd was 8.7 µg/g cr. (Table 8). Furthermore, the relationship between Cd exposure with BMD was not observed in Swedish individuals with low Cd exposure levels [19,20] or in a Cd-polluted area in Poland; however, Cd exposure levels were markedly lower (urinary Cd was 1.08 µg/g cr. in females and 0.88 µg/g cr. in males) [21] than those in Cd-polluted areas in the present study. On the other hand, Chinese individuals with high environmental Cd exposure showed that urinary Cd was associated with BMD and other markers of bone metabolism, along with decreased renal tubular function [22]. These findings are all in line with the spectrum.

This discrepancy may be attributed to differences in subjects, such as races and lifestyles. For example, Swedish women generally have more vulnerable bones [23] than female Japanese farmers, who perform markedly heavier loads of farming work. This discrepancy cannot be explained solely by biological or environmental factors. We speculate that it is mainly derived from technical aspects of data handling, such as the interpretation of a significant *p*-value as an indicator. It is not always appropriate to judge “significance” based only on a *p*-value, often *p* < 0.05, because *p*-values decrease as the number of subjects increases, resulting in a “false positive” result [24]. A typical example is a correlation coefficient. In studies with a large number of subjects, BMD “correlated” with Cd exposure because *p*-values for the significance of correlation coefficients were less than 0.05; however, actual correlation coefficients were less than 0.2, indicating that the relationship was low. For example, in a previous study, Spearman’s rank correlation coefficient between urinary Cd and BMD was −0.12 (*p* ≤ 0.001) in 820 Swedish women, showing no relationship visually in a scatter diagram; however, the authors reported that “urinary Cd was negatively associated with BMD” [25]. Therefore, when the number of subjects is large, the judgment of significance needs to rely on the statistical values themselves, not *p*-values.

Another issue associated with the interpretation of data is the presentation of the results of a multivariate analysis. In many studies that reported a relationship between decreases in BMD and Cd exposure, the findings of multivariable analyses were not sufficiently presented: only regression coefficients not adjusted by SE or their *p*-values were shown. In a linear multiple regression analysis with a large number of subjects, it is appropriate to present the standard regression coefficients of all independent variables that are adjusted by SE and compare the values themselves in order to judge their contributions to BMD. Since age and weight affect BMD, the presentation of their standard regression coefficients as independent variables is essential, and comparisons with those of Cd exposure are also important. We herein showed that the standard regression coefficients of age and weight were markedly higher than those of blood or urinary Cd in multiple regression analyses (Table 11), clearly indicating that the effects of Cd on BMD were negligible relative to those of age and weight. Therefore, the apparent, but very subtle, effect of Cd on bones, which often disappears after adjustments for confounding factors, may have been over-evaluated in the majority of studies that reported a relationship between Cd and BMD. It is important to adjust and interpret the contribution of Cd to BMD based on comparative observations of Cd and other factors.

Nevertheless, Cd may have a direct, although very slight, effect on bones, because previous studies suggested the relevance of smoking in BMD [20,26]. When Cd enters the body through smoking, it is absorbed through lungs into the bloodstream and reaches bone tissues directly, while Cd in food absorbed from the intestines travels to the liver through the portal vein and induces the production of metallothionein, a low-molecular-weight protein that is rich in cysteine and binds to Cd to detoxify it [27], and then circulates around the body. Therefore, in the case of Cd exposure by smoking, highly reactive ionic Cd that does not bind to metallothionein may directly affect bones, particularly when the level of Cd that accumulates in the body is so low that metallothionein production is minimal. Cd has also been shown to inhibit osteoblast activities and stimulate osteoclast formation in vitro [28]. Therefore, ionic Cd derived from smoking may directly affect bones; however, its effects are weak and temporal and are not comparable to the severe bone injuries induced secondarily through renal tubular dysfunction, such as osteomalacia in *itai-itai* disease.

The present study has a number of limitations that need to be addressed. Since this was a cross-sectional study, the causal relationship between Cd exposure and BMD remains unclear. However, the half-life of Cd that accumulates in the body is very long (10–30 years), enabling us to consider the design of this study as a type of longitudinal analysis because internal Cd exposure continues without marked changes for decades. We adjusted the urinary concentrations of Cd, α1MG, and ß2MG using that of Cr, which may have underestimated their concentrations in elderly or female subjects, in whom muscle volumes and the urinary excretion of Cr are small. In area A, no significant differences were observed in blood Cd between age classes; however, urinary Cd levels decrease with age, suggesting the effects of the adjustment by urinary Cr. The classification of subjects by age and comparisons within the same age classes minimized this effect and revealed significant differences in age-dependent increases in urinary Cd, α1MG, and ß2MG between the three areas. Furthermore, multiple regression analyses including urinary Cr as an independent variable excluded the effects of urinary Cr adjustments and clearly showed the net relationships of urinary substances with BMD. The subjects with diabetes, whose BMD might have been affected, were not excluded from the analyses. It is difficult to distinguish such subjects clearly because blood sugar levels tend to fluctuate depending on therapy or lifestyle. However, we performed multiple regression analyses of BMD, adding HbA1c to the independent variables, showing no statistically significant relation for HbA1c.

## 5. Conclusions

We investigated the relationship between environmental Cd exposure and BMD in female farmers in two Cd-polluted areas and a control area in the northern part of Japan. A significant deterioration of renal tubular function and a small decrease in BMD were observed in older postmenopausal subjects exposed to a high level of Cd. However, multiple regression analyses of all subjects revealed no relationship between Cd and BMD. These results indicate that Cd exposure itself does not directly affect bones, supporting the long-standing theory, the spectrum of chronic Cd toxicity, where Cd induces bone injury, osteomalacia, secondarily from renal tubular dysfunction.

## Figures and Tables

**Figure 1 toxics-13-00688-f001:**
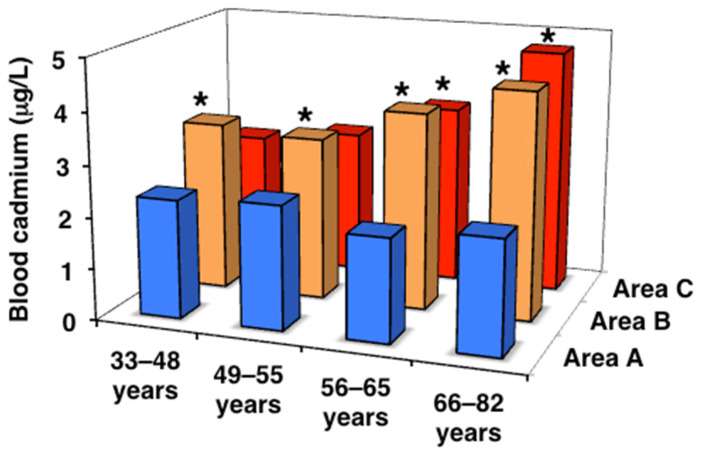
Cadmium concentrations in peripheral blood in the three areas. Data are presented as the median. * Significant difference from the value in area A.

**Figure 2 toxics-13-00688-f002:**
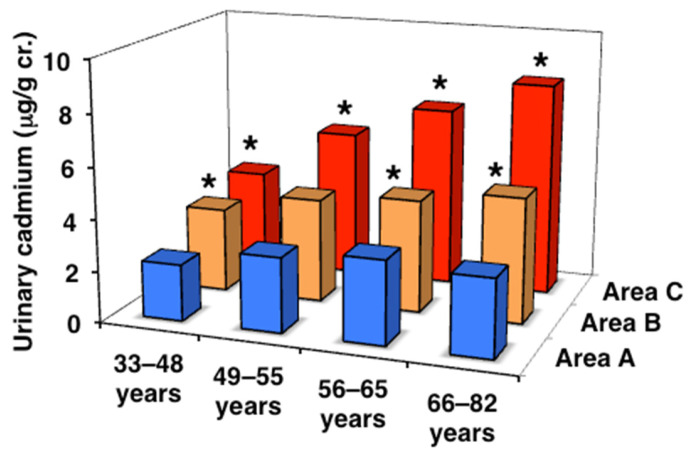
Cadmium concentrations in urine in the three areas. Data are presented as the median. * Significant difference from the value in area A.

**Figure 3 toxics-13-00688-f003:**
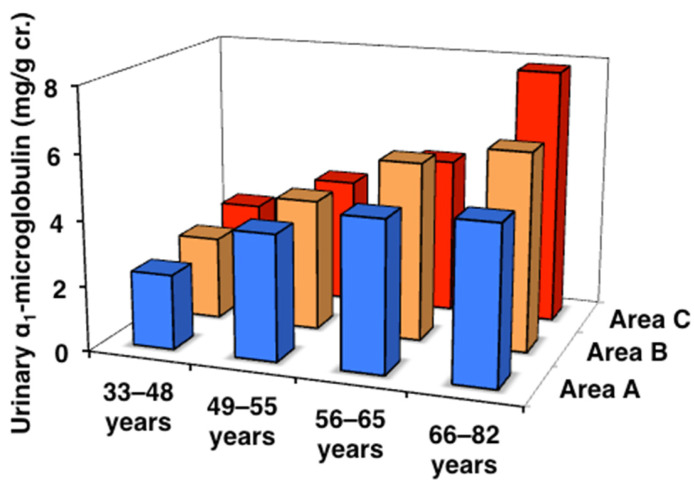
Urinary α1-microglobulin concentrations in the three areas. Data are presented as the median.

**Figure 4 toxics-13-00688-f004:**
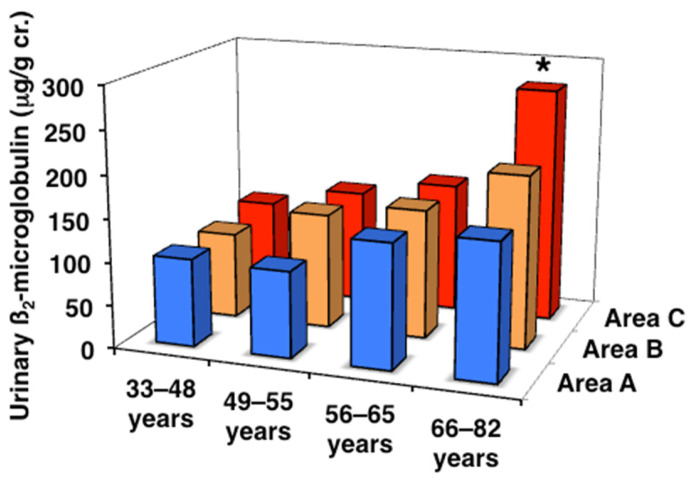
Urinary β2-microglobulin concentrations in the three areas. Data are presented as the median. * Significant difference from the value in area A.

**Figure 5 toxics-13-00688-f005:**
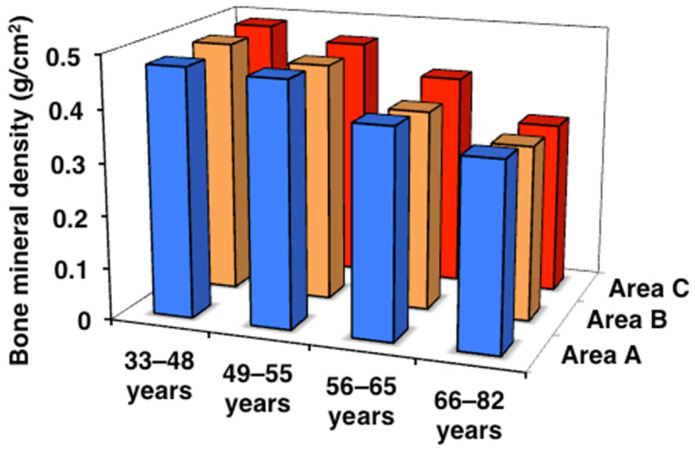
Bone mineral density in the three areas. Data are presented as the median. Data are presented as the mean.

**Table 1 toxics-13-00688-t001:** Grouping of subjects from the three areas by age and menstrual status.

	Area A	Area B	Area C	Total
All ages				
Number	220	655	392	1267
Age	61.8 ± 7.7	57.6 ± 10.7 *	57.6 ± 9.0 *	58.3 ± 9.8
Age range	33–79	20–78	34–82	20–82
Serum LH	17.3 (11.9–21.9)	21.0 * (13.4–30.4)	22.1 * (13.5–30.3)	19.9 (13.0–28.8)
20–32 years				
Number	0	22	0	22
Age	-	27.3 ± 3.8	-	27.3 ± 3.8
Serum LH	-	4.7 (2.5–18.6)	-	4.7 (2.5–18.6)
33–48 years (premenopause)				
Number	13	99	63	175
Age	44.5 ± 3.8	43.6 ± 3.9	43.8 ± 3.7	43.7 ± 3.8
Serum LH	2.9 (1.7–5.4)	5.0 (3.2–9.5)	4.6 (2.8–8.9)	4.6 (2.8–8.5)
49–55 years (perimenopause)				
Number	32	136	101	269
Age	52.3 ± 2.1	52.3 ± 1.9	52.2 ± 2.1	52.3 ± 2.0
Serum LH	19.5 (10.9–26.3) #	27.2 (18.4–37.1) #	22.4 (13.7–37.0) #	24.5 (15.7–36.6) #
56–65 years (younger postmenopause)				
Number	97	228	149	474
Age	61.2 ± 2.9	61.1 ± 2.8	60.5 ± 2.6	60.9 ± 2.8
Serum LH	18.0 (13.7–22.4) #	23.9 (18.1–32.9) *#	26.7 (19.1–34.1) *#	23.2 (17.3–31.1) #
66–82 years (older postmenopause)				
Number	78	170	79	327
Age	69.3 ± 3.0	69.1 ± 3.0	69.9 ± 3.5	69.4 ± 3.1
Serum LH	17.1 (12.1–19.9) #	20.7 (16.1–27.2) *#	22.1 (16.9–26.3) *#	19.6 (15.2–2 5.9) #

Data on age are presented as the arithmetic mean ± SD, while data on serum luteinizing hormone (LH) (mIU/mL) are shown as the median (25–75th percentile). * Significant difference from age and LH values in area A (*p* < 0.05 using the Bonferroni test or Steel–Dwass test). # Significant difference from LH values in the 33–48 years class (*p* < 0.05 using the Steel–Dwass test).

**Table 2 toxics-13-00688-t002:** Age-classified cadmium concentrations in peripheral blood and urine in the three areas (total number = 1245).

	Area A	Area B	Area C
Number	220	633	392
Blood cadmium (µg/L)			
All ages	2.10 (1.60–2.88)	3.78 (2.71–5.09) *	3.39 (2.38–5.14) *
	Range 0.76–6.90	Range 0.55–13.1	Range 0.74–31.2
33–48 years	2.30 (1.25–3.20)	3.29 (2.20–4.89)	2.57 (1.79–4.13)
49–55 years	2.40 (1.73–3.10)	3.16 (2.39–4.62) *	2.80 (2.07–4.20)
56–65 years	2.00 (1.50–2.70)	3.82 (2.84–4.97) *	3.47 (2.50–5.16) *
66–82 years	2.20 (1.80–2.80)	4.39 (3.35–5.45) *#	4.73 (3.35–6.66) *#
J-T test	*p* = 0.560	*p* < 0.001	*p* < 0.001
Urinary cadmium (µg/g cr.)			
All ages	3.02 (2.35–4.32)	4.29 (3.00–6.03) *	6.15 (4.29–8.76) *†
	Range 0.90–16.7	Range 0.51–27.3	Range 0.35–29.7
33–48 years	2.19 (1.48–2.68)	3.26 (2.35–4.85) *	3.75 (2.37–5.05) *
49–55 years	2.90 (2.05–4.86)	4.02 (2.84–5.96)	5.72 (3.99–7.24) *†#
56–65 years	3.24 (2.48–4.38) #	4.37 (3.29–6.18) *#	6.99 (4.90–9.16) *†#
66–82 years	3.01 (2.41–4.33) #	4.85 (3.53–6.61) *#	8.27 (6.14–10.5) *†#
J-T test	*p* = 0.358	*p* < 0.001	*p* < 0.001

Data are presented as the median (25th–75th percentile). * Significant difference from the value in area A (*p* < 0.05 using the Steel–Dwass test). † Significant difference from the value in area B (*p* < 0.05 using the Steel–Dwass test). # Significant difference from the value in the 33–48 years class (*p* < 0.05 using the Steel–Dwass test). J-T test: the Jonckheere–Terpstra trend test.

**Table 3 toxics-13-00688-t003:** Age-classified renal function in the three areas (total number = 1245).

	Area A	Area B	Area C
Number	220	633	392
Urinary α_1_-microglobulin (mg/g cr.)			
All ages	4.43 (2.74–7.13)	4.72 (3.00–8.05)	4.51 (2.63–7.39)
	Range ND–24.1	Range ND–56.0	Range ND–48.6
33–48 years	2.30 (1.41–3.24)	2.54 (1.84–4.18)	2.80 (2.05–5.20)
49–55 years	3.87 (2.51–6.22)	4.04 (2.57–6.89) #	3.87 (2.34–5.60)
56–65 years	4.63 (2.72–6.86)	5.49 (3.76–8.74) #	4.81 (2.85–7.47) #
66–82 years	4.82 (3.44–9.43) #	6.08 (3.94–10.9) #	7.85 (4.51–12.5) #
J-T test	*p* = 0.001	*p* < 0.001	*p* < 0.001
Urinary β_2_-microglobulin (μg/g cr.)			
All ages	133 (83–210)	146 (94–249)	146 (95–278)
	Range ND–1220	Range ND–5690	Range ND–15,300
33–48 years	102 (94–118)	100 (75–143)	109 (70–155)
49–55 years	100 (78–168)	134 (95–228) #	132 (80–202)
56–65 years	144 (81–213)	150 (107–230) #	151 (97–273) #
66–82 years	157 (83–275)	200 (119–408) #	273 (139–539) *#
J-T test	*p* = 0.016	*p* < 0.001	*p* < 0.001
eGFR (mL/min/1.73 m^2^)			
All ages	79.7 ± 14.8	82.0 ± 14.7	78.5 ± 14.2 †
	Range 30.6–130.0	Range 43.5–143.5	Range 28.2–133.0
33–48 years	90.0 ± 13.0	88.3 ± 14.8	86.3 ± 14.6
49–55 years	82.3 ± 13.1	84.4 ± 13.5	81.3 ± 13.3
56–65 years	80.9 ± 14.4	82.5 ± 13.8 #	76.9 ± 12.8 #
66–82 years	75.3 ± 15.0 #	75.7 ± 14.4 #	72.0 ± 14.2 #
J-T test	*p* < 0.001	*p* < 0.001	*p* < 0.001

Data are presented as the median (25th–75th percentile) or the mean ± SD. ND: not detected (α_1_-microglobulin, <0.9 mg/L; β_2_-microglobulin, <70 μg/L). * Significant difference from the value in area A (*p* < 0.05 using the Steel–Dwass test). † Significant difference from the value in area B (*p* < 0.05 using the Steel–Dwass test). # Significant difference from the value in the 33–48 years class (*p* < 0.05 using the Steel–Dwass test or Bonferroni test). J-T test: the Jonckheere–Terpstra trend test.

**Table 4 toxics-13-00688-t004:** Age-classified body weight, body mass index (BMI), and grip strength in the three areas (total number = 1245).

	Area A	Area B	Area C
Number	220	633	392
Body weight (kg)			
All ages	56.9 ± 8.1	55.6 ± 8.2	56.4 ± 8.9
	Range 36.7–83.4	Range 34.0–92.6	Range 34.2–100.3
33–48 years	58.0 ± 7.1	58.0 ± 8.9	57.3 ± 9.4
49–55 years	59.1 ± 7.4	57.0 ± 8.0	58.8 ± 8.9
56–65 years	57.6 ± 8.3	55.3 ± 7.8	56.6 ± 8.6
66–82 years	54.8 ± 8.0	53.6 ± 7.9 #	52.3 ± 8.1 #
BMI			
All ages	24.4 ± 3.2	24.2 ± 3.2	24.1 ± 3.4
	Range 17.8–35.4	Range 16.5–37.2	Range 16.3–43.0
33–48 years	23.4 ± 2.8	23.8 ± 3.4	23.3 ± 4.0
49–55 years	24.1 ± 2.5	24.1 ± 3.1	24.6 ± 3.2
56–65 years	24.7 ± 3.2	24.4 ± 3.1	24.5 ± 3.4
66–82 years	24.5 ± 3.4	24.3 ± 3.2	23.5 ± 2.8
Grip strength (kg)			
All ages	25.5 ± 4.1	24.8 ± 4.8	26.0 ± 4.6 †
	Range 14.5–36.0	Range 6.0–39.5	Range 10.0–38.5
33–48 years	28.3 ± 2.8	27.6 ± 4.3	29.9 ± 4.7 †
49–55 years	27.8 ± 3.9	26.2 ± 4.1	26.9 ± 3.8 #
56–65 years	25.7 ± 4.0	24.8 ± 4.6 #	25.4 ± 3.9 #
66–82 years	23.8 ± 3.7 #	22.0 ± 4.4 *#	23.0 ± 4.5 #

Data are presented as the mean ± SD. * Significant difference from the value in area A (*p* < 0.05 using the Bonferroni test). † Significant difference from the value in area B (*p* < 0.05 using the Bonferroni test). # Significant difference from the value in the 33–48 years class (*p* < 0.05 using the Bonferroni test).

**Table 5 toxics-13-00688-t005:** Age-classified bone mineral density and YAM% in the three areas (total number = 1245).

	Area A	Area B	Area C
Number	220	633	392
Bone mineral density (g/cm^2^)			
All ages	0.397 ± 0.082	0.400 ± 0.087	0.418 ± 0.089 *†
	Range 0.141–0.624	Range 0.185–0.680	Range 0.207–0.641
33–48 years	0.475 ± 0.044	0.483 ± 0.055	0.487 ± 0.058
49–55 years	0.466 ± 0.067	0.455 ± 0.071 #	0.462 ± 0.066
56–65 years	0.397 ± 0.073 #	0.381 ± 0.070 #	0.406 ± 0.078 †#
66–82 years	0.355 ± 0.074 #	0.332 ± 0.065 #	0.328 ± 0.071 #
J-T test	*p* < 0.001	*p* < 0.001	*p* < 0.001
YAM% (%)			
All ages	82.5 ± 17.0	83.4 ± 18.5	86.9 ± 18.5 *†
	Range 29.2–129.7	Range 39.0–188.0	Range 43.1–133.2
33–48 years	98.8 ± 9.1	100.6 ± 11.5	101.3 ± 12.0
49–55 years	96.9 ± 14.0	95.4 ± 16.6	96.0 ± 13.7
56–65 years	82.6 ± 15.2 #	79.4 ± 14.5 #	84.7 ± 16.2 †#
66–82 years	73.8 ± 15.3 #	69.1 ± 13.5 #	68.2 ± 14.7 #
J-T test	*p* < 0.001	*p* < 0.001	*p* < 0.001

Data are presented as the mean ± SD. * Significant difference from the value in area A (*p* < 0.05 using the Bonferroni test). † Significant difference from the value in area B (*p* < 0.05 using the Bonferroni test). # Significant difference from the value of 33–48 years class (*p* < 0.05 using the Bonferroni test). J-T test: the Jonckheere–Terpstra trend test.

**Table 6 toxics-13-00688-t006:** Age-classified serum calcium and phosphate and urinary calcium in the three areas (total number = 1245).

	Area A	Area B	Area C
Number	220	633	392
Serum calcium (mg/dL)			
All ages	9.6 ± 0.3	9.4 ± 0.3 *	9.5 ± 0.3 *
	Range 8.9–10.6	Range 8.6–11.4	Range 8.5–10.5
33–48 years	9.5 ± 0.4	9.2 ± 0.3	9.4 ± 0.4
49–55 years	9.7 ± 0.3	9.5 ± 0.3 *#	9.5 ± 0.3 *
56–65 years	9.6 ± 0.3	9.5 ± 0.3 #	9.5 ± 0.4 #
66–82 years	9.6 ± 0.3	9.5 ± 0.3 #	9.5 ± 0.3 *
Serum phosphate (mg/dL)			
All ages	3.8 ± 0.4	3.4 ± 0.4 *	3.6 ± 0.4 *†
	Range 2.8–5.1	Range 2.2–4.6	Range 2.3–4.9
33–48 years	3.6 ± 0.4	3.2 ± 0.3 *	3.4 ± 0.4 †
49–55 years	3.8 ± 0.5	3.5 ± 0.4 *#	3.7 ± 0.4 †#
56–65 years	3.8 ± 0.5	3.5 ± 0.4 *#	3.6 ± 0.4 †
66–82 years	3.7 ± 0.4	3.5 ± 0.4 *#	3.6 ± 0.5
Urinary calcium (mg/g cr.)			
All ages	141.5 (100.4–222)	140.1 (95.2–205.8)	142.8 (100.3–204.4)
	20.2–549.1	9.7–471.4	22.9–500.4
33–48 years	99.8 (67.6–125.5)	100.1 (66.8–146.5)	100.2 (74.9–145.5)
49–55 years	150.0 (126.8–216.5)	161.4 (110.1–206.9) #	153.3 (106.5–204.0) #
56–65 years	149.2 (99.8–217.7)	153.5 (103.5–218.2) #	148.6 (108.5–218.5) #
66–82 years	140.2 (104.3–243.6)	156.5 (101.9–221.6) #	157.6 (114.1–236.5) #

Data are presented as the mean ± SD or as the median (25th–75th percentile). * Significant difference from the value in area A (*p* < 0.05 using the Bonferroni test). † Significant difference from the value in area B (*p* < 0.05 using the Bonferroni test). # Significant difference from the value in the 33–48 years class (*p* < 0.05 using the Bonferroni test or Steel–Dwass test).

**Table 7 toxics-13-00688-t007:** Age-classified bone metabolism markers in the three areas (total number = 1245).

	Area A	Area B	Area C
Number	220	633	392
Bone alkaline phosphatase (U/L)			
All ages	32.8 (24.9–40.4)	30.0 (22.8–37.1) *	32.3 (23.1–42.1) †
	Range 10.3–86.7	Range 9.1–81.1	Range 10.1–103
33–48 years	18.9 (16.3–25.5)	19.2 (15.7–23.5)	19.8 (15.9–25.7)
49–55 years	30.9 (22.5–38.9) #	29.4 (22.9–36.2) #	33.7 (24.0–46.8) #
56–65 years	36.2 (27.1–43.4) #	31.5 (25.8–38.0) #	35.9 (28.1–44.0) #
66–82 years	32.3 (24.9–40.1) #	35.7 (27.7–42.9) #	32.3 (23.6–41.3) #
Osteocalcin (ng/mL)			
All ages	8.6 (6.5–10.3)	8.4 (6.3–10.8)	7.8 (5.9–10.2)
	Range 2.6–21.4	Range 1.5–22.8	Range 2.1–20.7
33–48 years	5.4 (4.6–6.2)	5.6 (4.3–6.6)	4.9 (4.0–5.9)
49–55 years	7.5 (5.6–9.3)	8.6 (6.6–10.8) #	9.0 (5.5–10.7) #
56–65 years	8.9 (7.2–10.7) #	9.1 (7.4–11.2) #	8.4 (6.7–10.3) #
66–82 years	9.0 (7.0–10.4) #	9.3 (7.3–11.3) #	8.7 (7.0–10.8) #
Urinary NTx (nmol/mmol cr.)			
All ages	50.8 (37.6–66.3)	62.7 (43.1–82.0) *	49.0 (32.6–73.4) †
	Range 11.4–153	Range 13.1–220	Range 9.7–270
33–48 years	29.5 (22.9–43.9)	33.7 (26.9–44.8)	33.0 (23.4–44.6)
49–55 years	52.3 (31.9–64.8)	70.2 (44.5–84.3) *#	52.0 (30.8–74.3) †#
56–65 years	53.3 (42.6–67.3) #	68.3 (52.5–86.1) *#	50.5 (37.5–78.5) †#
66–82 years	51.8 (36.0–68.5) #	68.5 (51.0–85.4) *#	64.7 (40.9–86.4) #
Urinary deoxypyridinoline (nmol/mmol cr.)			
All ages	7.0 (6.0–8.2)	6.6 (5.1–8.1) *	6.8 (5.5–8.0)
	Range 2.8–13.5	Range 1.3–23.5	Range 3.0–29.0
33–48 years	6.0 (4.6–6.7)	5.8 (4.7–6.8)	5.7 (4.6–6.6)
49–55 years	7.1 (6.5–8.4)	7.1 (5.3–8.8) #	7.5 (5.9–8.7) #
56–65 years	7.1 (6.1–8.3)	7.0 (5.1–8.1) #	6.8 (5.5–7.9) #
66–82 years	7.0 (6.0–8.1)	6.6 (5.4–8.4) #	7.3 (6.2–8.5) #

Data are presented as the median (25th–75th percentile). * Significant difference from the value in area A (*p* < 0.05 using the Steel–Dwass test). † Significant difference from the value in area B (*p* < 0.05 using the Steel–Dwass test). # Significant difference from the value in the 33–48 years class (*p* < 0.05 using the Steel–Dwass test).

**Table 8 toxics-13-00688-t008:** Age- and urinary cadmium-classified cadmium concentrations in peripheral blood and urine in the three areas (total number = 1245).

	Urinary Cadmium (µg/g cr.)				J-T Test
	<3.0	≥3.0, <4.5	≥4.5, <6.5	≥6.5	
Number					
All ages	313	308	301	323	
33–48 years	78	45	33	19	
49–55 years	69	61	73	66	
56–65 years	98	129	108	139	
66–82 years	68	73	87	99	
Age					
All ages	56.1 ± 10.1	58.8 ± 8.7 *	59.4 ± 8.5 *	61.0 ± 7.8 *	
	Range 33–77	Range 36–79	Range 36–82	Range 34–77	
33–48 years	43.1 ± 4.2	44.0 ± 3.3	44.4 ± 3.4	44.6 ± 3.9	
49–55 years	51.8 ± 2.0	52.4 ± 2.0	52.6 ± 1.8	52.3 ± 2.2	
56–65 years	60.2 ± 2.7	61.2 ± 2.8	60.8 ± 2.9	61.3 ± 2.7	
66–82 years	69.7 ± 3.2	69.2 ± 3.2	69.2 ± 3.1	69.4 ± 3.2	
Blood cadmium (µg/L)	2.61 (1.64–2.30)	3.22 (2.10–2.95) *	3.78 (2.58–3.50) *	5.53 (3.51–4.90) *	*p* < 0.001
All ages	Range 0.74–9.86	Range 0.55–9.48	Range 1.08–15.0	Range 1.20–31.2	
	2.19 (1.51–3.21)	3.08 (2.09–4.10)	4.06 (3.13–5.16)	6.50 (4.73–7.56)	
33–48 years	2.40 (1.70–3.15)	2.72 (1.98–3.70)	2.88 (2.38–4.10) *	4.31 (2.99–6.11) *	*p* < 0.001
49–55 years	2.31 (1.50–3.23)	3.02 (2.12–3.88)	3.29 (2.47–4.39) *	4.83 (3.34–6.47) *	*p* < 0.001
56–65 years	2.30 (1.80–3.59)	2.97 (2.15–4.47) *	3.90 (2.98–5.22) *	5.27 (3.95–7.85) *	*p* < 0.001
66–82 years	2.61 (1.64–2.30)	3.22 (2.10–2.95)	3.78 (2.58–3.50) *	5.53 (3.51–4.90) *	*p* < 0.001
J-T test	*p* = 0.191	*p* = 0.270	*p* = 0.058	*p* = 0.055	
Urinary cadmium (µg/g cr.)	2.35 (1.89–2.67)	3.81 (3.39–4.14) *	5.36 (4.92–5.94) *	8.71 (7.40–10.32) *	*p* < 0.001
All ages	Range 0.35–3.00	Range 3.01–4.50	Range 4.50–6.50	Range 6.50–29.7	
	2.21 (1.82–2.55)	3.65 (3.28–4.20)	5.12 (4.75–5.73)	8.11 (7.21–9.34)	
33–48 years	2.40 (1.72–2.70)	3.86 (3.39–4.04) *	5.36 (4.94–5.91) *	7.92 (7.11–9.20) *	*p* < 0.001
49–55 years	2.36 (1.91–2.69)	3.93 (3.42–4.19) *	5.42 (4.92–5.87) *	8.91 (7.44–10.4) *	*p* < 0.001
56–65 years	2.41 (2.11–2.74)	3.73 (3.47–4.15) *	5.46 (4.93–5.99) *	9.00 (7.51–10.7) *	*p* < 0.001
66–82 years	2.35 (1.89–2.67)	3.81 (3.39–4.14) *	5.36 (4.92–5.94) *	8.71 (7.40–10.32) *	*p* < 0.001
J-T test	*p* = 0.012	*p* = 0.487	*p* = 0.041	*p* = 0.011	

Data are presented as the mean ± SD or as the median (25th–75th percentile). * Significant difference from the <3.0 urinary cadmium class (*p* < 0.05 using the Bonferroni test or Steel–Dwass test). J-T test: the Jonckheere–Terpstra trend test.

**Table 9 toxics-13-00688-t009:** Age- and urinary cadmium-classified urinary α_1_-microglobulin and β_2_-microglobulin, bone mineral density, and YAM% in the three areas (total number = 1245).

	Urinary Cadmium (µg/g cr.)				J-T Test
	<3.0	≥3.0, <4.5	≥4.5, <6.5	≥6.5	
Urinary α_1_-microglobulin (mg/g cr.)					
All ages	3.87 (2.41–6.40)	4.62 (2.77–7.32) *	4.76 (2.99–7.63) *	5.62 (3.36–8.87) *	*p* < 0.001
	Range ND–30.7	Range ND–28.1	Range ND–28.9	Range ND–56.0	
33–48 years	2.66 (1.76–3.79)	2.67 (2.03–4.72)	2.92 (2.05–5.87)	2.22 (1.70–3.86)	*p* = 0.344
49–55 years	3.48 (2.27–5.42)	3.87 (2.26–6.24)	4.02 (3.09–6.05)	4.62 (2.62–7.98)	*p* = 0.013
56–65 years	4.48 (3.00–7.10) #	5.15 (3.52–7.65) #	4.79 (2.76–7.92)	5.73 (3.55–8.17) #	*p* = 0.135
66–82 years	5.09 (3.57–10.1) #	5.72 (3.49–11.7) #	5.86 (4.06–9.60) #	7.21 (4.36–11.9) #	*p* = 0.039
J-T test	*p* < 0.001	*p* < 0.001	*p* < 0.001	*p* < 0.001	
Urinary β_2_-microglobulin (μg/g cr.)					
All ages	121.5 (84.3–193.8)	137.8 (83.3–206.1)	147.3 (93.0–250.5) *	180.0 (112.6–335.8) *	*p* < 0.001
	Range ND–1236	Range ND–2808	Range ND–9087	Range ND–15330	
33–48 years	100.6 (80.8–132.1)	103.6 (67.9–147.6)	115.1 (65.0–201.7)	101.3 (55.6–146.4)	*p* = 0.925
49–55 years	119.1 (83.1–185.6)	124.1 (75.4–161.8)	127.5 (87.5–220.7)	164.9 (91.3–295.5)	*p* = 0.013
56–65 years	142.6 (91.9–195.8) #	143.8 (89.6–212.3)	147.4 (90.5–232.1)	160.3 (112.0–286.9) #	*p* = 0.005
66–82 years	160.4 (89.7–301.5) #	166.2 (89.7–346.8)	200.7 (121.1–318.6) #	272.6 (137.0–650.3) #	*p* < 0.001
J-T test	*p* < 0.001	*p* < 0.001	*p* < 0.001	*p* < 0.001	
Bone mineral density (g/cm^2^)					
All ages	0.426 ± 0.086	0.406 ± 0.084 *	0.402 ± 0.084 *	0.386 ± 0.090 *	*p* < 0.001
	Range 0.141–0.631	Range 0.200–0.615	Range 0.231–0.680	Range 0.207–0.641	
33–48 years	0.484 ± 0.058	0.483 ± 0.050	0.476 ± 0.046	0.459 ± 0.071	*p* = 0.679
49–55 years	0.474 ± 0.061	0.460 ± 0.067	0.467 ± 0.073	0.465 ± 0.072	*p* = 0.647
56–65 years	0.386 ±0.081 #	0.380 ± 0.073 #	0.396 ± 0.062 #	0.380 ± 0.077 #	*p* = 0.932
66–82 years	0.360 ± 0.067 #	0.339 ± 0.074 #	0.332 ± 0.066 #	0.324 ± 0.068 #	*p* = 0.002
J-T test	*p* < 0.001	*p* < 0.001	*p* < 0.001	*p* < 0.001	
YAM%					
All ages	88.7 ± 17.8	84.4 ± 17.5 *	83.7 ± 17.5 *	80.7 ± 19.5 *	*p* < 0.001
	Range 29.2–131.2	Range 42.0–128.0	Range 48.0–142.0	Range 43.1–188.0	
33–48 years	101.0 ± 12.1	100.4 ± 10.5	99.0 ± 9.4	95.5 ± 14.8	*p* = 0.655
49–55 years	99.0 ± 12.8	95.6 ± 13.9	97.0 ± 15.2	97.0 ± 18.5	*p* = 0.644
56–65 years	80.2 ± 16.9 #	79.0 ± 15.3 #	82.4 ± 12.9 #	79.0 ± 16.1 #	*p* = 0.865
66–82 years	74.9 ± 13.9 #	70.9 ± 15.4 #	69.0 ± 13.8 #	67.3 ± 14.1 #	*p* = 0.002
J-T test	*p* < 0.001	*p* < 0.001	*p* < 0.001	*p* < 0.001	

Data are presented as the median (25th–75th percentile) or as the mean ± SD. ND: not detected (α_1_-microglobulin, <0.9 mg/L; β_2_-microglobulin, <70 μg/L). * Significant difference from the <3.0 urinary cadmium class (*p* < 0.05 using the Bonferroni test or Steel–Dwass test). # Significant difference from the 33–48 years class (*p* < 0.05 using the Bonferroni test or Steel–Dwass test). J-T test: the Jonckheere–Terpstra trend test.

**Table 10 toxics-13-00688-t010:** Multiple regression analyses of renal tubular function (total number = 1267).

Dependent Variables	Independent Variables	Regression Coefficients	β	*p*-Values	VIF
log urinary α_1_-microglobulin					
Model 1	Age	0.011	0.315	<0.001	1.1
(R′ = 0.340)	log urinary creatinine	0.761	0.546	<0.001	1.0
	log blood cadmium	0.112	0.074	0.002	1.1
Model 2	Age	0.011	0.311	<0.001	1.1
(R′ = 0.342)	log urinary creatinine	0.659	0.474	<0.001	1.9
	log urinary cadmium	0.119	0.109	<0.001	1.8
log urinary β_2_-microglobulin					
Model 1	Age	0.008	0.216	<0.001	1.1
(R′ = 0.215).	log urinary creatinine	0.591	0.406	<0.001	1.0
	log blood cadmium	0.262	0.166	<0.001	1.1
Model 2	Age	0.009	0.224	<0.001	1.1
(R′ = 0.202)	log urinary creatinine	0.441	0.303	<0.001	1.9
	log urinary cadmium	0.178	0.156	<0.001	1.8

R′: Multiple correlation coefficient adjusted for the degrees of freedom. β: Standard regression coefficient. VIF: variance inflation factor.

**Table 11 toxics-13-00688-t011:** Multiple regression analyses of bone mineral density (BMD) (total number = 1267).

Dependent Variables	Independent Variables	Regression Coefficients	β	*p*-Values	VIF
BMD, Model 1	Age	−0.004	−0.461	<0.001	1.4
(R′ = 0.472).	Body weight	0.003	0.318	<0.001	1.2
	Grip strength	0.002	0.116	<0.001	1.3
	log urinary creatinine	0.009	0.025	0.311	1.5
	log urinary α_1_-microglobulin	−0.014	−0.058	0.021	1.5
	log blood cadmium	−0.007	−0.018	0.386	1.1
Model 2	Age	−0.004	−0.469	<0.001	1.4
(R′ = 0.472).	Body weight	0.003	0.321	<0.001	1.2
	Grip strength	0.002	0.117	<0.001	1.3
	log urinary creatinine	0.001	0.004	0.893	2.2
	log urinary α_1_-microglobulin	−0.015	−0.063	0.013	1.5
	log urinary cadmium	0.009	0.035	0.213	1.8
Model 3	Age	−0.004	−0.463	<0.001	1.3
(R′ = 0.474)	Body weight	0.003	0.320	<0.001	1.2
	Grip strength	0.002	0.115	<0.001	1.3
	log urinary creatinine	0.008	0.024	0.297	1.3
	log urinary β_2_-microglobulin	−0.017	−0.075	0.001	1.3
	log blood cadmium	−0.004	−0.010	0.635	1.1
Model 4	Age	−0.004	−0.471	<0.001	1.3
(R′ = 0.475)	Body weight	0.003	0.323	<0.001	1.2
	Grip strength	0.002	0.115	<0.001	1.3
	log urinary creatinine	−0.001	−0.001	0.959	2.0
	log urinary β_2_-microglobulin	−0.019	−0.081	<0.001	1.3
	log urinary cadmium	0.011	0.041	0.145	1.9

R′: Multiple correlation coefficient adjusted for the degrees of freedom. β: Standard regression coefficient. VIF: Variance inflation factor.

## Data Availability

The raw data supporting the conclusions of this article will be made available by the authors on request.

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
