# Peer review of "Effects of Environmental Cadmium Exposure Sufficient to Induce Renal Tubular Dysfunction on Bone Mineral Density Among Female Farmers in Cadmium-Polluted Areas in Northern Japan"

_toxics, 2025, doi:10.3390/toxics13080688_

Round 1
Reviewer 1 Report
Comments and Suggestions for Authors
The study is aimed at identifying the health characteristics of women living in areas with cadmium contamination. The authors describe the research methodology in sufficient detail and carefully, as well as the results. I believe that the study is complete and can be published, as it provides new data on the effects of different levels of cadmium on the health of women of different age groups. The authors also conducted a thorough analysis of the literature on this topic, and provided their thoughts in the Discussion section. It can be recommended that researchers pay attention to the forms of metal and the ways in which cadmium enters the human body. Perhaps this is also one of the reasons for the difference in the threshold of exposure to cadmium among the population of different countries. Using the example of mercury, when the form of the element's location and routes of entry play an important role in further effects on the body. It's one thing to inhale fumes of inorganic mercury, another is to consume metal in organic form with food. I would also recommend that the authors first clearly outline the purpose and objectives of the study in the introduction, and of course expand the conclusions - since for such a significant material, the conclusions are quite compact - you can expand, identify the novelty of the study, what are the main differences from the studies of other authors, what recommendations for future experiments the authors give.
I also think we can recommend that some of the tabular material be presented in the form of graphs to visualize trends. And also to emphasize the importance of the findings for doctors and managers, for example.
Author Response
# We appreciate the positive comments.
# The reviewer recommends to pay attention to the forms of metal and the ways in which Cd enters the human body. However, although mercury is well known to show different routes of absorption and cause different health effects between metal mercury, inorganic mercury, and organic mercury, we are afraid that it is generally accepted that Cd does not have such differences between chemical forms. However, we discussed the difference of the effects of Cd on bones between Cd absorption via a gastrointestinal tract through food and Cd absorption via lungs through smoking in Discussion.
# In the Introduction, to clearly outline the purpose of this study, we changed “We also discussed intrinsic differences in the effects of various Cd exposure levels on bones.” into “in order to clarify intrinsic differences in the effects of various Cd exposure levels on bones.”.
# To identify the novelty of the study and emphasize the importance of the findings, we added in Conclusions “, supporting the long-standing theory, the spectrum of chronic Cd toxicity, where Cd induces bone injury, osteomalacia, secondarily from renal tubular dysfunction.”
# For the recommendations for future experiments, we already stated in Discussion “when the number of subjects is large, the judgment of significance needs to rely on the statistical values themselves, not p values.” and “It is important to adjust and interpret the contribution of Cd to BMD based on comparative observations of Cd and other factors.”.
# We added graphs of blood and urinary Cd, urinaryα1MG and ß2MG, and BMD as Figs. 1-5 in the manuscript.
Reviewer 2 Report
Comments and Suggestions for Authors
This manuscript examines the relationship between cadmium exposure and mean tubular dysfunction in three populations in Japan with varying environmental cadmium exposures. These associations were contrasted with effects of cadmium exposure on bone mineral density. The study is an expansion on an earlier study that came to difference conclusions that similar studies in other parts of the world. The authors confirm there earlier results and provide rational for differences between studies as the current results indicate no direct of environmental Cd exposure on bone. The study appears to be well performed. The manuscript is well written and easy to follow. Data is presented in a straightforward manner.
Major: There is no mention of obtaining informed written consent from the participants.
Minor: Please avoid the use of first person voice (e.g., "we") in the manuscript except in the introduction and conclusion. The manuscript will read much better.
Line 40 - comma after "environment"
Line 72 - second comma should be semicolon
Define "microgram/gram cr." at first use
Author Response
# We appreciate the positive comments.
# The reviewer mentioned that there is no mention of obtaining informed written consent from the participants. But, we stated in the “2.2. Procedures for health examinations” as “and obtained written informed consent from all subjects”.
# We changed the active voices, “We …”, into passive voices in Materials and Methods and in Results.
# We added comma after "environment" at the first line of Introduction.
# We changed comma after "in the northern part of Japan [4, 5]" into semicolon in Introduction.
# We added “, and presented like “µg/g cr.” ” in “2.5. Statistical analysis”.
Reviewer 3 Report
Comments and Suggestions for Authors
This present study sought to investigate whether cadmium (Cd) exposure, at concentrations sufficient to impair renal tubular function, influences bone mineral density (BMD) in female farmers living in Cd-polluted regions of northern Japan. The findings can be directly applied to clinical screening approaches for populations exposed to heavy metals, emphasizing early kidney monitoring over a sole focus on bone density decline.
My comments:
- ABSTRACT:
1.1 Lines 16 – 18: “We then expanded JMETS to the most Cd-polluted area in northern Japan, Akita prefecture, with area A as the control and areas B and C as Cd-polluted areas (Cd exposure levels: B < C)……………….”.
The authors need to mention Cd concentration levels in areas A, B and C, and also note whether statistical significance was observed (p – Value).
1.2 Lines 21 – 23: “The distribution of blood and urinary Cd levels was areas A < B < C, with the steepest age-dependent increase in area C, particularly in older post menopausal subjects with a urinary Cd level around the threshold for renal tubular dysfunction”.
The authors need to mention the concentration levels of Cd, and also note whether statistical significance was observed (p – Value).
1.3 Lines 23 – 33: Same comments as above mention concentration levels of parameters and also note whether statistical significance was observed (p – Value).
- INTRODUCTION:
2.1 Presented adequately – covers key aspects related to the research project.
- MATERIALS AND METHODS:
3.1 Lines 106 – 109: “Exclusion criteria were as follows: ex-and current smokers, the presence of chronic renal failure, nephritis, renal tumors, rheumatoid arthritis, systemic lupus erythematosus, sarcoidosis, an insufficient volume of blood sampled, incomplete answers in questionnaires, and less than a 10-year history of eating locally harvested rice.”
Low BMD is also associated with Osteoporosis, Hyperthyroidism and Diabetes mellitus. Did the authors collect this data? Medications such as Corticosteroids (e.g., prednisone) can lead to significant bone loss with long-term use as well.
3.2 Apart for Cd, was other confounding exposures also assessed such as particulate matter?
- RESULTS:
4.1 The results are presented and explained well.
4.2 Table 1 needs to be neatened, and numerical values must be aligned to each parameter – same for all other tables in manuscript. Also, unit of measurement for age and Serum LH is missing.
- DISCUSSION AND CONCLUSION:
- Adequately written in line with the aim and objectives of this study.
Overall, I approve publication once the above minor comments are addressed.
Author Response
# We appreciate the positive comments.
# The reviewer requested to mention Cd concentrations of the areas in Abstract, but we are afraid that there are not such data in this manuscript.
# We added “(blood Cd: 2.10, 3.78, 3.39 µg/L, and urinary Cd: 3.02, 4.29, 6.15 µg/g cr., respectively, p < 0.05)” and “(273 and 157 μg/g cr., respectively, p < 0.05)” in Abstract.
# With regard to exclusion criteria, we also excluded subjects with hyperthyroidism and steroid hormone therapy, which were overlapped with other criteria. Therefore, we added “hyperthyroidism, steroid hormone therapy” in “2.1. Study areas and populations”. The reviewer also mentioned osteoporosis as an exclusion criteria, but it should not be excluded from the analyses because the subjects with lowered BMD, which we measured in all subjects, should be included, given the aim of this study.
# Some of the subjects in this study had a history of diabetes, and showed relatively high levels of HbA1c. However, the blood sugar levels tend to be fluctuant depending on therapy or lifestyle, and it is not easy to exclude such subjects from the analyses. Actually, we performed multiple regression analyses of BMD adding HbA1c to the independent variables, showing no statistically significant relation for HbA1c. Therefore, we added “The subjects with diabetes, whose BMD might have been affected, were not excluded from the analyses. It is difficult to distinguish such subjects clearly because blood sugar levels tend to be fluctuant depending on therapy or lifestyle. However, we performed multiple regression analyses of BMD adding HbA1c to the independent variables, showing no statistical significance for HbA1c.” in Discussion.
# We did not investigate exposure to particulate matters in the study subjects. I am afraid that particulate matters would not be related with BMD.
# We are afraid that the numerical values are aligned to each parameter in all tables.
# We added LH unit, (mIU/mL), in Table 1.